# Foscarnet Versus Ganciclovir for Severe Congenital Cytomegalovirus Infection: Short- and Long-Term Follow-Up

**DOI:** 10.3390/v17050720

**Published:** 2025-05-17

**Authors:** Giovanni Nigro, Marta Buzzi, Milena Catenaro, Eleonora Coclite, Mario Muselli

**Affiliations:** 1Non-Profit Association Mother-Infant Cytomegalovirus Infection (AMICI), 00042 Rome, Italy; 2Azienda Sanitaria Locale, 66100 Chieti, Italy; 3Pediatric Department, Santo Spirito Hospital, 65121 Pescara, Italy; 4Azienda Sanitaria Locale, 64100 Teramo, Italy; 5Department of Life, Health and Environmental Sciences, University of L’Aquila, 67100 L’Aquila, Italy

**Keywords:** congenital cytomegalovirus infection, encephalopathy, sensorineural hearing loss, ganciclovir, foscarnet

## Abstract

Background: Cytomegalovirus (CMV) infection is the most common and serious congenital infection, with universal screening in pregnancy, standardized therapy, and a vaccine still lacking. Study design: In the 1990s, we noted that intravenous ganciclovir did not cure some children with severe sequelae due to congenital cytomegalovirus (CMV) infection. Therefore, we performed an open randomized trial using intravenous foscarnet as an alternative to intravenous ganciclovir in 24 infants (12 in each therapy group), all with severe neurological manifestations due to congenital CMV infection. Nine and five infants, belonging to the foscarnet or ganciclovir group, respectively, had abnormal hearing. One infant in each group also had chorioretinitis. Concomitantly, 12 CMV-infected infants with similar manifestations, who did not receive any therapy, were used as controls. The results of short-term (2 years) and long-term (7–29 years, mean 22.2) follow-up are reported herein. Short-term results: Neurological outcomes were normal in five of the twelve children who were treated with foscarnet, compared to nine of the twelve children given ganciclovir. None of the untreated children were healthy. There was a statistically significant difference (*p* = 0.023) between the treated and untreated children. Hearing was normal in four of the twelve children treated with foscarnet, seven of the twelve children treated with ganciclovir, and two untreated children. Long-term-results: Two children in both therapy groups died before the age of 17 years, and six untreated children died between 7 and 26 years of age. Neurological outcomes were normal in three of the ten children treated with foscarnet, in two of the ten treated with ganciclovir, and in none of the untreated children. Hearing was normal in two children treated with foscarnet, in six children treated with ganciclovir, and in one untreated child. Conclusions: Intravenous ganciclovir and foscarnet were found to be safe at long-term follow-up and appeared to be capable of mitigating the neurological and auditory consequences of congenital CMV disease at the short-term follow-up. However, there was progressive worsening of the symptomatology in all three groups, with a statistically significant increase in the number of deaths (*p* = 0.035) among 4 of the 24 children in the therapy groups and 6 of the 12 untreated children.

## 1. Introduction

Cytomegalovirus (CMV) infection is the most common and serious congenital infection occurring in approximately 0.5–2% of all neonates, depending on the socio-economic levels of the population [1]. Overall, congenital CMV infection is symptomatic in about 10% of infected neonates and associated with significant neurological and sensorineural sequelae in about half of them. Furthermore, 8–13% of asymptomatic CMV-infected neonates will subsequently develop neurological, visual, or auditory defects [2,3]. When transmitted in the first three months from a pregnant woman with primary infection, CMV can cause severe neurological and sensorineural sequelae in up to 50% of cases because of the high susceptibility of embryonal cells to CMV and the lack of maternal and fetal immune defenses against the virus [4,5,6]. Both symptomatic and asymptomatic children with congenital CMV infection will excrete the virus in their urine or saliva for years, with it being a potential source of infection, particularly for non-immune pregnant women [7].

Ganciclovir, if given orally for several months to years, could decrease the frequency and severity of neurological sequelae and sensorineural hearing loss (SNHL) [8,9]. Unfortunately, in the 1990s, only one drug—intravenous ganciclovir—was studied in congenital CMV infection. After ganciclovir was administered for two weeks, the outcomes of infants with severe symptomatic congenital CMV infection were poor [10,11]. Foscarnet is a non-nucleoside pyrophosphate analog used alternatively or in combination with ganciclovir for patients with AIDS and CMV retinitis [12,13]. We performed an open randomized controlled study with intravenous foscarnet alternated with intravenous ganciclovir in 24 infants. Untreated controls included 12 CMV-infected infants with similar symptoms.

## 2. Methods

*Pharmacological properties of the drugs:* Ganciclovir is an acyclic analog of the nucleoside guanosine, which requires phosphorylation to achieve antiviral activity. After penetrating the CMV-infected cell, ganciclovir is converted to ganciclovir-5′-monophosphate by a viral kinase, phosphotransferase, which is encoded by the CMV gene UL97. Subsequently, ganciclovir–diphosphate and ganciclovir–triphosphate are catalyzed by cellular phosphatases, which are 10-fold more concentrated in CMV-infected cells than in uninfected cells. Ganciclovir–triphosphate takes the place of guanosine–triphosphate in the viral DNA polymerase, serving as a poor substrate for chain elongation, thereby disrupting viral DNA synthesis [14].

In 1995, foscarnet was the only anti-herpes drug that was not a nucleoside analog, since it inhibits the activity of the viral DNA polymerase by binding the pyrophosphate site and blocking the cleavage of pyrophosphate from the terminal nucleoside triphosphate added to the growing DNA chain. Because foscarnet does not require phosphorylation by kinases, it is active against pUL 97 kinase variations that confer resistance to ganciclovir [15].

*Study design:* From 15 September 1995 to 30 June 1999, an open, controlled, randomized study was performed to compare foscarnet therapy with ganciclovir therapy in infants with multisystem CMV involvement, including neurological abnormalities. A third group of children with symptomatic CMV disease, who had been brought to our attention too late to be included in one of the therapy groups, were followed up as an untreated group. Based on serological CMV results in pregnancy, 22 infants were born to mothers who had a primary infection (14 mothers showed CMV seroconversion and 8 had high IgM titers and low IgG-avidity percentages) in the first four months of pregnancy. The mothers of the remaining 14 infants were IgG positive but IgM negative to CMV at their first antibody screening within 11 weeks of pregnancy; apart from two mothers who showed a subsequent > 2-fold increase in their IgG titers with high avidity values in the second test, the remaining mothers had no further tests performed during pregnancy. Therefore, it is presumable that a non-primary (reactivation or reinfection) CMV infection occurred in their first weeks of pregnancy, given the severe symptomatology of the neonates.

After careful presentation to the parents of the pharmacological properties, mechanism of activity, and reported side effects, a signed informed consent form was obtained. All treated infants were enrolled within one month of birth. The endpoints of the study were as follows: (1) inhibition of CMV replication and (2) improvement of symptoms.

*Diagnostic and clinical features:* Congenital CMV infection was diagnosed through CMV isolation and/or polymerase chain reaction (PCR) from the infants’ urine within three weeks of age. Neurological involvement was defined by microcephaly, periventricular calcifications, cerebral atrophy or hypotrophy, leukodystrophy, cortical malformations such as polymicrogyria and lissencephaly, seizures associated with positive CMV DNA in the cerebrospinal fluid (CSF), and ventricular abnormalities. Sensorineural involvement mostly included deafness or hypoacusia. A few infants also had chorioretinitis.

CMV disease at ≥two years was defined by mental and/or motor retardation and persistent auditory or visual impairment. Mental retardation was defined by an IQ standard score of <70. The diagnostic scale systems used were Stanford–Binet and Bayley III. Normal hearing was defined by 0–20 dB thresholds. Abnormal responses were rated as mild (21–45 dB), moderate (46–70 dB), and severe (≥71 dB) thresholds.

*Therapy courses and dosage:* As shown in Table 1, Table 2 and Table 3, our study included 3 groups of 12 patients (9 males in each group), which were divided into the following groups: (1) the Foscarnet Group; (2) the Ganciclovir Group; and (3) the Control Group.

Foscarnet Group: 12 infants received intravenous foscarnet (Foscavir^®^) at a dosage of 60 mg/kg body weight every 8 h for 2 weeks and a dosage of 90 mg/kg body weight given three times a week for three months as maintenance therapy. Ganciclovir Group: 12 infants received ganciclovir (Cytovene^®^) at a dose of 7.5 mg/kg body weight every 8 h for 2 weeks and 10 mg/kg three times weekly for three months. The dosage protocol was based on the pharmacological and therapeutic properties of both drugs and previous experiences [16,17]. Control Group: 12 infants who were at non-study hospitals and who were not enrolled in spite of severe symptomatic congenital CMV infection.

Instrumental examinations included ophthalmoscopy, cerebral and abdominal ultrasound, brainstem evoked responses (BSERs), cerebral CT-scan, and/or MRI. In neonates with seizures, cerebrospinal fluid (CSF) was drawn, and an electroencephalogram was also performed.

*Virological investigations:* CMV isolation was performed with shell vial procedures using human diploid fibroblast cells. Cell monolayers were fixed 24–48 h after inoculation and were stained using a direct immunofluorescence test using an anti-CMV immediate early antigen fluorescein-conjugated monoclonal antibody (E13 clone Argene, Varilhes, France). CMV DNA was detected in neonatal serum, urine, saliva, and cerebrospinal fluid, using nested PCR. For the determination of CMV genomic copies, competitive QPCR was performed using CMV-ibridoquant from Amplimedical-Bioline (Turin, Italy), according to the manufacturer’s instructions. Detection of CMV-specific IgG and IgM antibodies was performed using an enzyme immunoassay with a diagnostic kit from Radim (Pomezia, Italy). IgG avidity to CMV was measured with a kit from Bouty (Sesto San Giovanni, Italy).

### Statistical Analysis

Variables are expressed as the number of cases and percentages. The chi-squared test was used for comparisons between the treatment groups, and an alpha level of 0.05 was considered to be statistically significant. The statistical analysis was performed using STATA 18 software for Windows.

## 3. Results

*Symptomatology at birth:* As shown in Table 1 and Table 2, all 24 infants treated with antiviral drugs had severe neurological abnormalities.

*Short-term results:* Neurological outcomes were normal in five children who were treated with foscarnet, in three given ganciclovir, and in none of the untreated children. Statistically, antiviral therapy, both drugs vs. no therapy (*p* = 0.047) and therapy vs. no therapy (*p* = 0.023), significantly improved their neurological symptoms (Table 4 and Table 5). As a single therapy, only foscarnet therapy was statistically significant (*p* = 0.012) (Table 6).

Hearing was normal in four children treated with foscarnet, seven treated with ganciclovir, and two untreated children. Bilateral deafness occurred in one infant treated with foscarnet, two who had ganciclovir, and in four untreated children. Statistical analysis showed that hearing was only improved significantly (*p* = 0.035) by ganciclovir therapy (Table 7). Visual impairment occurred in one infant in each therapy group and in two untreated children.

*Long-term-results:* The long-term follow-up was similar in both the children treated with foscarnet (mean 24.2 years, ranging from 15 to 29), those given ganciclovir (mean 21.4, ranging from 11 to 28), and in children who did not receive any therapy (mean 20.5, ranging from 7 to 27). Neurological outcomes were normal in three children treated with foscarnet, in two treated with ganciclovir, and in none of the untreated children was normal in two children treated with foscarnet, six treated with ganciclovir, and one untreated child (Figure 1). Statistical analysis did not significant differences among the three groups of children (*p* = 0.197). Hearing statistical analysis showed that the antiviral therapy, as both drugs vs. no therapy, significantly (*p* = 0.045) improved the infants’ hearing (Table 4). In particular, only ganciclovir therapy was statistically significant (*p* = 0.025) (Table 7) (Figure 2).

Visual impairment persisted in one infant in each therapy group, while blindness developed in two untreated children. Two children in both therapy groups died before the age of 17 years, and six untreated children died between 7 and 28 years of age. Statistical analysis showed that the antiviral therapy, as both drugs vs. no therapy, significantly (*p* = 0.03) decreased mortality (Table 5).

*Adverse effects*: The only significant adverse event was the development of hyporegenerative anemia, which occurred in five infants, four of whom were treated with foscarnet and one was treated with ganciclovir. Anemia regressed following therapy with erythropoietin (EPREX^®^) in three infants treated with foscarnet but required transient suspension of antiviral therapy in the fourth infant who was affected by β-microcythemia. No other side effects, including renal toxicity, occurred.

*Virological outcomes:* After the first two-week treatment, CMV DNA detection was negative in the urine of three infants (25%) treated with foscarnet and that of two (16.7%) treated with ganciclovir. After the three-month phase of therapy, all but one (treated with ganciclovir) infant stopped excreting CMV DNA in their urine. Ganciclovir resistance was not tested.

## 4. Discussion

The results of our study provide evidence that antiviral therapy could be efficient in achieving a short-term improvement or lack of worsening in both neurological and hearing abnormalities caused by congenital CMV infection. Compared to untreated children, both antiviral drugs appeared to be capable of temporarily mitigating the consequences of CMV infection. As previously reported in children treated with ganciclovir for a 3½-month regimen, a few children treated with foscarnet using the same regimen protocol also had favorable short-term outcomes [16]. Although there was not a statistically significant difference in the outcomes between the two drugs, the short-term neurological results obtained with foscarnet were slightly better than those achieved with ganciclovir. On the contrary, possibly due to a lower initial number of hearing abnormalities, the long-term hearing results were statistically better than those obtained with foscarnet. However, the long-term follow-up revealed the development of neurological abnormalities and worsening hearing in both therapy groups, and two children in each group died. The unfavorable outcome was due to the reactivated replication of CMV in the infected cells after short-term inhibition induced by both drugs and the consequent persistent viral activity for many years. Only two children treated with foscarnet and one who was treated with ganciclovir showed normal development. Following the therapy courses in this study, we reported on the favorable outcome of foscarnet therapy for an infant with hepatic fibrosis due to congenital CMV infection [18]. On the other hand, an unfavorable outcome occurred in all untreated children, even at the short-term follow-up, and half of them died during the long-term follow-up.

Concerning the possible development of side effects, foscarnet caused hyporegenerative anemia more frequently than ganciclovir during the therapy courses. The long-term follow-up did not reveal drug-related adverse events, including tumors, which have been observed in some animals treated with ganciclovir. Therefore, our study shows that both foscarnet and ganciclovir are safe, and foscarnet could be an efficient therapeutic alternative to ganciclovir for short therapy courses. In fact, for long-term treatments, oral formulations of ganciclovir make this drug preferable to foscarnet, which can only be given intravenously.

In conclusion, there are two relevant findings from our study: (1) The infants with severe congenital CMV infection should be treated to stop viral replication and consequent worsening of the symptomatology. This result may be obtained only if the diagnosis of congenital CMV infection is performed soon after birth, even better in pregnancy. In fact, two preventive approaches for congenital CMV disease are currently available, valaciclovir and hyperimmune globulin, which could be associated given their different mechanisms of activity [19,20]. (2) Antiviral therapy for children with cerebral and/or sensorineural lesions, currently using oral valganciclovir, should be given as long as possible because CMV is excreted in the urine and saliva for many years after birth, presumably also replicating in neuronal and cochlear cells, which are not accessible for CMV detection. Since ganciclovir is only active after triple phosphorylation promoted by CMV, the dosage should be gradually decreased after the negative detection of CMV DNA in the urine, avoiding the rebound effects of high CMV reactivation after abruptly stopping valganciclovir given for six months at increasing dosages related to the increasing weight of the infants [9].

## Figures and Tables

**Figure 1 viruses-17-00720-f001:**
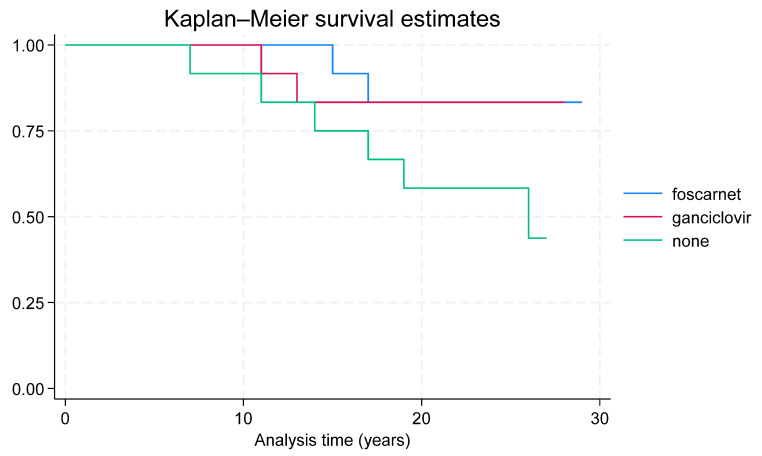
Long-term survival of treated and untreated children.

**Figure 2 viruses-17-00720-f002:**
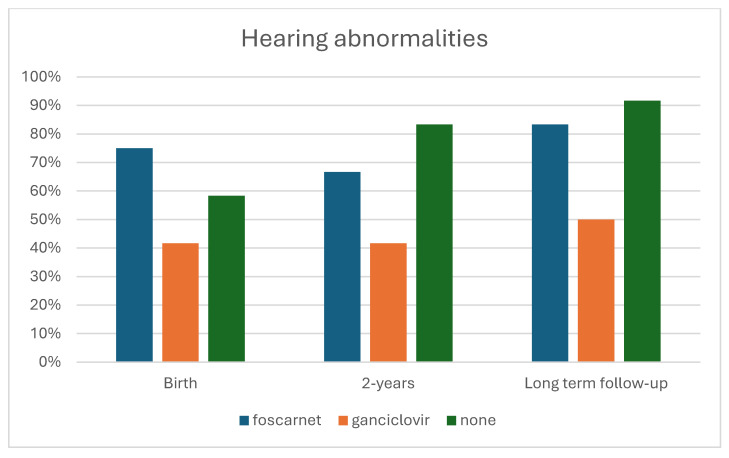
Cumulative incidence of hearing abnormalities by treatment.

**Table 1 viruses-17-00720-t001:** Findings from 12 children with severe congenital CMV infection who were treated with foscarnet.

Patient Sex Maternal Infection Birth Date	Main Neurological Abnormalities	Hearing	Short-Term Outcomes (2 Years)	Long-Term Outcomes (Years of Follow-Up)
1. M Primary 13.03.99	Cerebral hypoplasia Ventriculomegaly Leukoencephalopathy	Right deafness	Right deafness Left hypoacusia	Mental retardation Bilateral deafness (23)
2. M Non-primary 19.03.97	Cerebral hypoplasia Ventriculomegaly Leukoencephalopathy	Normal	Normal	Normal (26)
3. F Primary 13.02.96	Microcephaly Cerebral hypoplasia Ventriculomegaly Leukoencephalopathy	Right hypoacusia	Mental motor retardation Right hypoacusia	Mental motor retardation Right deafness (28)
4. F Non-primary 29.09.95	Microcephaly Calcifications Lissencephaly Ventriculomegaly	Left hypoacusia	Mental retardation Tetraparesis Left hypoacusia	Severe mental retardation Tetraparesis Left deafness (27)
5. M Primary 12.09.95	Microcephaly Ventriculomegaly Cerebral and cerebellar hypoplasia	Left deafness	Mental retardation Bilateral deafness	Mental retardation Bilateral deafness (28)
6. M Primary 20.04.96	Microcephaly Ventriculomegaly Calcifications Schizencephaly	Bilateral deafness	Severe mental retardation Tetraplegia Bilateral deafness	Severe mental retardation Tetraplegia Bilateral deafness Death (15)
7. M Non-primary 14.11.95	Cerebral hypoplasia Ventriculomegaly Leukoencephalopathy	Right hypoacusia	Normal	Normal (29)
8. M Non-primary 31.09.96	Microcephaly Calcification Leukoencephalopathy	Normal	Mental retardation	Mental retardation Bilateral hypoacusis (27)
9. M Primary 15.03.98	Microcephaly Cerebral hypoplasia Ventriculomegaly Leukoencephalopathy	Bilateral hypoacusia	Bilateral hypoacusia	Mental retardation Bilateral deafness (21)
10. M Primary 06.09.98	Microcephaly Calcifications Lissencephaly Ventriculomegaly	Normal	Mental retardation Paraparesis	Tetraparesis Bilateral hypoacusia Death (17)
11. F Primary 25-11-97	Calcifications Ventriculomegaly Chorioretinitis	Right hypoacusia	Motor retardation Right hypoacusia Visual impairment	Mental motor retardation Bilateral hypoacusia Visual impairment (21)
12. M Non-primary 21.05.97	Cerebral hypoplasia Ventriculomegaly Leukoencephalopathy	Left hypoacusia	Left hypoacusia	Left deafness (28)

**Table 2 viruses-17-00720-t002:** Findings from 12 infants with congenital cerebral CMV infection treated with ganciclovir.

Patient Sex Maternal Infection Birth Date	Main Neurological Abnormalities	Hearing	Short-Term Outcomes (2 Years)	Long-Term Outcomes (Years of Follow-Up)
1. M Non-primary 2-02-98	Microcephaly Polymicroyria Ventriculomegaly	Bilateral hypoacusia	Mental retardation Bilateral hypoacusia	Mental retardation Bilateral hypoacusia (26)
2. M Primary 27-06-99	Calcifications Microcephaly Leukoencephalopathy	Normal	Mental retardation	Mental retardation Left hypoacusia (25)
3. M Primary 03-08-97	Microcephaly Calcifications Cerebral aplasia	Left deafness	Severe mental retardation Tetraparesis Left deafness	Severe mental retardation Tetraplegia Bilateral deafness Death (13)
4. M Non-primary 17-09-95	Microcephaly Calcifications Lissencephaly Hydrocephalus	Bilateral deafness	Severe mental retardation Tetraparesis Bilateral deafness	Tetraplegia Bilateral deafness Death (11)
5. F Primary 29.12.95	Microcephaly Calcifications Cerebral hypoplasia	Right hypoacusia	Mental motor retardation Right hypoacusia	Mental motor retardation Right deafness (28)
6. F Non-primary 30.03.97	Microcephaly Ventriculomegaly Cerebral hypoplasia	Normal	Normal	Normal (27)
7. M Non-primary 06.05.96	Microcephaly Calcifications Leukoencephalopathy	Normal	Mental motor retardation	Mental motor retardation (24)
8. M Non-primary 31-06-98	Cerebral hypoplasia Ventriculomegaly Leukoencephalopathy	Bilateral deafness	Bilateral deafness	Mental retardation Bilateral deafness (25)
9. M Primary 13.11.95	Ventriculomegaly Cerebral hypoplasia Chorioretinitis	Normal	Visual impairment	Mental retardation Visual impairment (26)
10. M Primary 26-03-96	Microcephaly Calcifications Ventriculomegaly	Normal	Mental motor retardation	Mental retardation Paraparesis (28)
11. F Primary 24-01-97	Cerebral hypoplasia Ventriculomegaly Hemiplegia	Normal	Right hemiparesis	Right hemiparesis (27)
12. M Primary 12-01-99	Microcephaly Calcifications Ventriculomegaly	Normal	Mental motor retardation	Mental motor retardation (24)

**Table 3 viruses-17-00720-t003:** Findings from 12 children with severe congenital neurological CMV infection who were not treated with antivirals.

Patient Sex Maternal Infection Birth Date	Main Neurological Abnormalities	Hearing	Short-Term Outcomes (2 Years)	Long-Term Outcomes (Years of Follow-Up)
1. F Primary 10.07.98	Leukoencephalopathy Ventriculomegaly Paraplegia	Normal	Mental retardation Right hypoacusia Paraplegia	Mental retardation Bilateral hypoacusia Tetraparesis (24)
2. M Non-primary 19-11-97	Microcephaly Cerebral and cerebellar hypoplasia	Bilateral hypoacusia	Mental motor retardation Bilateral deafness	Mental retardation Bilateral deafness (26)
3. M Non-primary 04.11.95	Hydrocephalus Cerebral hypoplasia Chorioretinitis	Normal	Severe mental motor retardation Left hypoacusia Visual impairment	Tetraplegia Blindness Bilateral hypoacusia Death (26)
4. M Primary 21.11.06	Cerebral and cerebellar hypoplasia Calcifications Hydrocephalus	Bilateral hypoacusia	Severe mental motor retardation Bilateral hypoacusia	Severe mental retardation Tetraparesis Bilateral deafness Death (14)
5. M Primary 12.05.97	Cerebral and cerebellar hypoplasia Calcifications Lissencephaly	Bilateral deafness	Severe mental motor retardation Bilateral deafness	Severe mental retardation Tetraplegia Bilateral deafness Death (7)
6. M Non-primary 12.05.98	Microcephaly Ventriculomegaly Tetraparesis	Normal	Mental retardation Tetraparesis	Severe mental retardation Tetraplegia Bilateral hypoacusia Death (11)
7. M Non-primary 21.09.97	Microcephaly Cerebral hypoplasia Calcifications Leukoencephalopathy	Bilateral hypoacusia	Severe mental motor retardation Bilateral hypoacusia	Tetraplegia Bilateral deafness Death (19)
8. F Primary 10.01.99	Microcephaly Calcifications Ventriculomegaly	Normal	Mental motor retardation Left hypoacusia	Bilateral hypoacusia Mental motor retardation (24)
9. M Primary 19.02.97	Microcephaly Cerebral atrophy Leukoencephalopathy	Bilateral hypoacusia	Mental motor retardation Bilateral deafness	Severe mental motor retardation Bilateral deafness (26)
10. F Primary 27-2-98	Cerebral atrophy Ventriculomegaly Leukoencephalopathy	Normal	Mental motor retardation	Mental retardation Paraparesis (25)
11. M Primary 27.02.96	Microcephaly Ventriculomegaly Leukoencephalopathy Chorioretinitis	Bilateral hypoacusia	Mental motor retardation Bilateral deafness Visual impairment	Mental retardation Bilateral deafness Blindness (27)
12. M Non-primary 11.06.96	Microcephaly Calcifications Paraplegia	Bilateral hypoacusia	Severe mental motor retardation Bilateral hypoacusia	Tetraparesis Bilateral deafness Death (17)

**Table 4 viruses-17-00720-t004:** Short-term and long-term outcomes in 12 children who were treated with foscarnet, 12 children treated with ganciclovir, and 12 untreated children.

	Foscarnet *n* = 12	Ganciclovir *n* = 12	No Therapy *n* = 12	*p*-Value
At birth				
Neurological abnormalities	12 (100.0%)	12 (100.0%)	12 (100.0%)	-
Hearing abnormalities	9 (75.0%)	5 (41.7%)	7 (58.3%)	0.254
Death	0 (0.0%)	0 (0.0%)	0 (0.0%)	-
Short-term outcomes (2 years)				
Neurological abnormalities	7 (58.3%)	9 (75.0%)	12 (100.0%)	0.047
Hearing abnormalities	8 (66.7%)	5 (41.7%)	10 (88.9%)	0.102
Death	0 (0.0%)	0 (0.0%)	0 (0.0%)	-
Long-term outcomes (24 years)				
Neurological abnormalities	9 (75.0%)	10 (88.9%)	12 (100.0%)	0.197
Hearing abnormalities	10 (88.9%)	6 (50.0%)	11 (91.7%)	**0.045**
Death	2 (16.7%)	2 (16.7%)	6 (50.0%)	0.109

**Table 5 viruses-17-00720-t005:** Short-term and long-term outcomes in 24 children who were treated with foscarnet or ganciclovir (Therapy), compared to 12 untreated children (No Therapy).

	Therapy *n* = 24	No Therapy *n* = 12	*p*-Value
At birth			
Neurological abnormalities	24 (100.0%)	12 (100.0%)	-
Hearing abnormalities	14 (58.3%)	7 (58.3%)	1.000
Death	0 (0.0%)	0 (0.0%)	-
Short-term outcomes (2 years)			
Neurological abnormalities	16 (66.7%)	12 (100.0%)	**0.023**
Hearing abnormalities	13 (54.2%)	10 (88.9%)	0.086
Death	0 (0.0%)	0 (0.0%)	-
Long-term outcomes (24 years)			
Neurological abnormalities	19 (79.2%)	12 (100.0%)	0.088
Hearing abnormalities	16 (66.7%)	11 (91.7%)	0.102
Death	4 (16.7%)	6 (50.0%)	**0.035**

**Table 6 viruses-17-00720-t006:** Short-term and long-term outcomes in 12 children who were treated with foscarnet, compared to 12 untreated children.

	Foscarnet *n* = 12	No Therapy *n* = 12	*p*-Value
At birth			
Neurological abnormalities	12 (100.0%)	12 (100.0%)	-
Hearing abnormalities	9 (75.0%)	7 (58.3%)	0.386
Death	0 (0.0%)	0 (0.0%)	-
Short-term outcomes (2 years)			
Neurological abnormalities	7 (58.3%)	12 (100.0%)	**0.012**
Hearing abnormalities	8 (66.7%)	10 (88.9%)	0.346
Death	0 (0.0%)	0 (0.0%)	-
Long-term outcomes (24 years)			
Neurological abnormalities	9 (75.0%)	12 (100.0%)	0.064
Hearing abnormalities	10 (88.9%)	11 (91.7%)	0.537
Death	2 (16.7%)	6 (50.0%)	0.083

**Table 7 viruses-17-00720-t007:** Short-term and long-term outcomes in 12 children who were treated with ganciclovir, compared to 12 untreated children.

	Ganciclovir *n* = 12	No Therapy *n* = 12	*p*-Value
At birth			
Neurological abnormalities	12 (100.0%)	12 (100.0%)	-
Hearing abnormalities	5 (41.7%)	7 (58.3%)	0.414
Death	0 (0.0%)	0 (0.0%)	-
Short-term outcomes (2 years)			
Neurological abnormalities	9 (75.0%)	12 (100.0%)	0.064
Hearing abnormalities	5 (41.7%)	10 (88.9%)	**0.035**
Death	0 (0.0%)	0 (0.0%)	-
Long-term outcomes (24 years)			
Neurological abnormalities	10 (88.9%)	12 (100.0%)	0.140
Hearing abnormalities	6 (50.0%)	11 (91.7%)	**0.025**
Death	2 (16.7%)	6 (50.0%)	0.083

## Data Availability

The data presented in this study are available from the corresponding author upon request.

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
