# Peer review of "Foscarnet Versus Ganciclovir for Severe Congenital Cytomegalovirus Infection: Short- and Long-Term Follow-Up"

_viruses, 2025, doi:10.3390/v17050720_

Round 1
Reviewer 1 Report
Comments and Suggestions for Authors
In the study submitted by Nigro et al, the authors describe the results of an open randomized trial which started for about 30 years. In this study, congenitally infected children with severe neurological manifestations were treated shortly after birth with iv Ganciclovir or iv Foscarnet or without any antiviral treatment. Due the long follow-up period the authors were able to assess not only the short-term results but also the long-term outcomes which makes this study outstanding. The authors could show that in overall treated children are more likely to improve hearing and neurological abnormalities than untreated children, however also many of the treated children showed a worsening of the symptoms during the long-term follow-up. This study is concise and well written, and the detailed description of the symptoms over such a long observation period makes this study very interesting.
Please find below my comments.
L18: I would also mention in the abstract that 24 infants means 12 in each treatment group.
L75: Letermovir and Maribavir are also anti-CMV drugs which are not a nucleoside analog, maybe this should be mentioned?
L86, L87: Has an IgG avidity test been done? This might strengthen the reactivation/reinfection conclusion.
L184-L195:
Table 4 and Table 5 show all the data. In my opinion it would be easier for the reader to comprise the statistical analyses in these two tables and skip the Tables 6-8.
It might be my misunderstanding but I think, there are some typos in the results text: L151: „both as“ can be removed; L153: Table 7 instead of 6?; L157: p=0,035 instead of 0,033?; L158: Table 8 instead of 7?; L174: p=0,035 instead of 0,03?
L183: has it been tested for ganciclovir resistance?
L209-L211:
I think, this is a very important discussion point, but I am missing the data. It might be very interesting to show that CMV starts to replicate (detectable CMV DNAemia) after the short-term inhibition.
Author Response
REFEREE 1
I wish to thank you for your comments, which will improve the paper.
Please find below my answers to your comments.
L18: I would also mention in the abstract that 24 infants means 12 in each treatment group.
A: (12 in each therapy group) added
L75: Letermovir and Maribavir are also anti-CMV drugs which are not a nucleoside analog, maybe this should be mentioned?
A: Since both drugs were not available when the trial was started (Maribavir was authorized by EU/Italy in 2007, Letermovir by FDA in 2017, the sentence was changed to “In 1995, foscarnet was…”
L86, L87: Has an IgG avidity test been done? This might strengthen the reactivation/reinfection conclusion.
A: Avidity test was performed only in the mothers with primary infection, as shown in the added sentence in parenthesis in L87-88 (14 mothers showed a CMV seroconversion, 8 had high IgM titers and low IgG-avidity percentages), and in the two mothers with increased IgG titers, as reported in L91.
L184-L195: Table 4 and Table 5 show all the data. In my opinion it would be easier for the reader to comprise the statistical analyses in these two tables and skip the Tables 6-8.
A: Table 6 was eliminated. Tables 7 and 8 are needed because of a few statistically significant results
It might be my misunderstanding but I think, there are some typos in the results text: L151: „both as“ can be removed; L153: Table 7 instead of 6?; L157: p=0,035 instead of 0,033?; L158: Table 8 instead of 7?; L174: p=0,035 instead of 0,03?
A: “both as” was removed; Table 6 was correctly noted by you as Table 7 (now 6); you are correct: p=0.035; you are right about Table 8 (which is now 7); yes 0.0.035
L183: has it been tested for ganciclovir resistance?
- Ganciclovir resistance was not tested.
L209-L211:
I think, this is a very important discussion point, but I am missing the data. It might be very interesting to show that CMV starts to replicate (detectable CMV DNAemia) after the short-term inhibition.
A: You are right, there is not detectable CMV DNAemia, but CMV DNA replication in the infected cells (added in L213)
Reviewer 2 Report
Comments and Suggestions for Authors
The text by Nigro et al. describes a series of 24 patients with congenital CMV infection treated in the 1990s with Foscarnet or Ganciclovir (along with a control group).
The authors document how early treatment with both drugs leads to some responses in neurological development and sensory damage with both Foscarnet and Ganciclovir.
The authors document the development of hyporegenerative anemia as the main side effect of treatment.
Problems:
1) the authors should specify the time of initiation of therapy (days of life of the patient),
2) the authors should report any renal toxicity and if absent state it in the text,
3) the authors should present a curve of incidence of sensory damage during the months/years of life of the patient,
4) the authors should report a curve of survival of the patients,
5) the authors should specify in the text the meaning of primary and non-primary infection in the mother
Author Response
REFEREE 2
Thank you of your comments
Problems:
1) the authors should specify the time of initiation of therapy (days of life of the patient),
A: In L94.95 we have reported: All treated infants were enrolled within one month from birth.
2) the authors should report any renal toxicity and if absent state it in the text,
A: In L 181-2 we added the following sentence: No other side effects, including renal toxicity, occurred.
3) the authors should present a curve of incidence of sensory damage during the months/years of life of the patient,
A: Figure 1 was added on page 12
4) the authors should report a curve of survival of the patients,
A: Figure 2 was added on page 12
5) the authors should specify in the text the meaning of primary and non-primary infection in the mothe
A: In L 93, we specified the meaning of non-primary CMV infection
Round 2
Reviewer 2 Report
Comments and Suggestions for Authors
The authors fully replied to all queries
Author Response
Thank you.